# Comparative Visual Outcome Analysis of a Diffractive Multifocal Intraocular Lens and a New Diffractive Multifocal Lens with Extended Depth of Focus

**DOI:** 10.3390/jcm11247374

**Published:** 2022-12-12

**Authors:** Majid Moshirfar, Seth R. Stapley, Wyatt M. Corbin, Nour Bundogji, Matthew Conley, Ines M. Darquea, Yasmyne C. Ronquillo, Phillip C. Hoopes

**Affiliations:** 1Hoopes Vision Research Center, Hoopes Vision, Draper, UT 84020, USA; 2John A. Moran Eye Center, Department of Ophthalmology and Visual Sciences, University of Utah, Salt Lake City, UT 84112, USA; 3Utah Lions Eye Bank, Murray, UT 84107, USA; 4Arizona College of Osteopathic Medicine, Midwestern University, Glendale, AZ 85308, USA; 5Stritch School of Medicine, Loyola University Chicago, Chicago, IL 60153, USA; 6School of Medicine, University of Utah, Salt Lake City, UT 84132, USA

**Keywords:** refractive outcomes, multifocal IOL, Synergy, PanOptix, IOL, visual acuity, photic phenomena, cataract, glare, halo, Symfony, extended depth-of-focus (EDOF)

## Abstract

This study compares the visual and refractive performance of the TECNIS Synergy (DFR00V-DFW150-225-300-375) multifocal intraocular lens (IOL) and the AcrySof IQ PanOptix (TFAT00-30-40-50-60) multifocal IOL. Patients who underwent phacoemulsification and cataract extraction and received either a multifocal Synergy or PanOptix IOL were included. Monocular uncorrected distance (UDVA), intermediate (UIVA), near (UNVA), and corrected distance (CDVA) visual acuities were assessed at three and six months postoperatively. Secondary outcome measures of photic phenomena were also assessed. A total of 140 patients (224 eyes) were included in this study, with 69 patients (105 eyes) in the Synergy group and 71 patients (119 eyes) in the PanOptix group. There were no statistically significant differences in UIVA or CDVA measurements across all time points. When assessing UDVA, at three months postoperatively, there were more eyes in the PanOptix group with vision better than 20/40 (*p* = 0.04). At three and six months postoperatively, the average UNVA was superior in the Synergy group (*p* = 0.01, 0.002). While the Synergy group reported more night vision disturbances at one and three months (*p* = 0.01, 0.03), the PanOptix group had more night vision disturbances at six months (*p* = 0.02). Although not statistically significant, the AcrySof IQ PanOptix multifocal IOL demonstrated better UDVA and UIVA sooner postoperatively than the TECNIS Synergy multifocal IOL. The Synergy IOL provided statistically better UNVA compared to the PanOptix IOL at three and six months postoperatively. Synergy patients reported more early photic phenomena than PanOptix patients, which later diminished.

## 1. Introduction

Cataracts contribute significantly to blindness worldwide and are the leading cause in middle- and low-income countries [1,2]. Traditionally, cataract surgery replaces the natural lens with an artificial monofocal intraocular lens (IOL) implant, focused either for distance or near vision. Recently, new multifocal and extended depth-of-focus (EDOF) IOL technologies have become available, allowing patients to have more spectacle independence following surgery [3]. While multifocal technology creates several focal points for distance, intermediate, and near vision, EDOF technology works by elongating the focal point and enhancing the range of vision or depth of focus [4]. This study will compare the visual and refractive performance of the TECNIS Synergy multifocal IOL (Synergy) and the AcrySof IQ PanOptix multifocal IOL (PanOptix).

The Synergy IOL, model DFR00V, was created by Johnson & Johnson Vision in Santa Ana, CA, and obtained its approval from the U.S. Food and Drug Administration (FDA) on 6 May 2021 [5]. This hydrophobic acrylic IOL filters violet light and absorbs ultraviolet (UV) light. The Synergy IOL is biconvex and has an anterior aspheric surface with achromatic technology designed to correct chromatic aberration and improve image contrast [5]. Its proprietary diffractive surface uses multifocal and extended depth of focus technologies, allowing for a greater range of vision [4].

The PanOptix IOL, model TFNT00/TFAT00, was first released in 2015 by Alcon Laboratories in Fort Worth, TX. Due to its intermediate focal point of 60 cm (arms-length), this one-piece lens is considered to have a more comfortable near-to-intermediate range of vision than other multifocal IOLs, which normally have an 80 cm intermediate focal point [6]. The PanOptix IOL is constructed from an acrylate/methylacrylate copolymer, contains two open-loop haptics, and utilizes a quadrifocal design [7]. The quadrifocal design provides diffraction orders at a distance, intermediate, and near ranges while adding an extra order for distance to improve distance visual acuity [6].

To date, there have been eight studies on Synergy IOL, two of which have reported six months of postoperative visual outcomes (Appendix A) [8,9,10]. The purpose of this study is to add to the current literature by comparing visual outcomes of the Synergy IOL to the PanOptix IOL at distance, intermediate, and near visual ranges over the span of six months from the date of surgery. Visual outcomes for both IOLs will be compared to existing published literature and FDA clinical trial approval outcomes.

## 2. Materials and Methods

This retrospective, comparative study used deidentified medical record data from 140 patients (224 eyes) operated on by two different surgeons at a tertiary surgery center. The surgeons in this study implanted both types of IOLs. All patients were fully informed and consented to phacoemulsification cataract surgery paired with the implantation of either a Synergy or a PanOptix IOL between 1 January 2021 and 1 March 2022. None of the eyes underwent femtosecond-laser-assisted cataract surgery. Exclusion criteria included patients with a history of glaucoma, previous corneal disease, retinal abnormalities, trauma to the eye, congenital ocular abnormalities, use of ocular medications with a possible effect on vision, perioperative or postoperative complications, cerebrovascular accidents affecting vision, and degenerative eye disorders. Preoperative refractive error did not exceed 10 diopters for myopic correction or 3 diopters for astigmatic correction. The rate of YAG capsulotomy was also monitored between the two groups. The Hoopes Vision Ethics Committee approved this study, which adheres to the tenets of the Declaration of Helsinki. This retrospective study was also approved by the Biomedical Research Alliance of New York (BRANY, Lake Success, NY, USA) (#-A20-12-547-823).

This study analyzed both primary and secondary visual outcomes within the two separate treatment groups based on IOL type. Primary outcomes included monocular-uncorrected distance visual acuity (UDVA), uncorrected intermediate visual acuity (UIVA) measured at 66 cm, near visual acuity (UNVA) measured at 40 cm, corrected distance visual acuity (CDVA), postoperative manifest refractive sphere (MRS), mean refractive cylinder (MRC), and mean refraction spherical equivalent (MRSE). These outcomes were assessed at three and six months postoperatively. Both UNVA and UIVA were measured on a Jaeger scale and converted to Snellen units using a conversion chart [11]. All Snellen units were then converted to logMAR using a standard conversion formula for calculation of average visual acuity and statistical analysis. The astigmatism double-angle plot tool provided by the American Society of Cataract and Refractive Surgery was used for the vector analysis. These data were presented and organized in accordance with the method proposed by Abulafia et al. [12].

Secondary outcomes included patient-reported photic phenomena, specifically glare, halo, night vision disturbances, photophobia, and dryness, which were recorded based on the subjective reporting of the patient at each postoperative visit (one-, three-, and six months).

### 2.1. Statistical Analysis

In the variables described above, summary descriptive statistics were calculated, and continuous variables were statistically analyzed using a one-way ANOVA F-test to determine unequal outcomes of variables. In outcomes where there was a statistically significant difference, further statistical analysis was performed using a two-tailed hypothesis unpaired *t*-test. All discrete variables were assessed using a chi-squared test. A *p*-value less than 0.05 was considered statistically significant. All statistical analyses were performed using Microsoft Excel (v. 2206, Microsoft Corp., Redmond, WA, USA).

### 2.2. Surgical Technique

Cataract extraction was performed under sterile conditions in the operating room. A keratome was used to make a clear corneal incision measuring 2.4 mm, and a continuous curvilinear capsulorhexis measuring 5.0–5.5 mm was performed. Phacoemulsification was performed using a horizontal chop or a divide-and-conquer fashion utilizing the Infiniti Vision System (Alcon Laboratories, Inc. Fort Worth, TX, USA). No complications occurred, and all the wounds were confirmed as self-sealing.

Patients were directed to use fluoroquinolone third- or fourth-generation antibiotic eye drops four times each day for one month. Patients began topical steroid eye drops four times each day and tapered weekly over the course of one month. Patients also began ketorolac (0.3%) eye drops twice daily for six weeks.

### 2.3. IOL Calculation

Preoperative biometry measurements (flat keratometry, steep keratometry, flat axis, steep axis, white-to-white, lens thickness, axial length, and aqueous chamber depth) were obtained before surgery utilizing the Lenstar LS 900 (Haag-Streit, Mason, OH, USA) and the Zeiss IOLMaster 700 (Carl Zeiss Meditec AG, Jena, Germany). The Barrett Universal II Formula, provided by the Asia-Pacific Association of Cataract and Refractive Surgeons, was used to obtain predicted postoperative refraction for both Synergy and PanOptix patients, including a target of emmetropia. In order to optimize near vision, IOL powers for the Synergy and PanOptix lenses were determined by selecting the lens which was predicted to produce a postoperative spherical equivalent of 0 to +0.25 D, per manufacturer recommendation. For Synergy, an A-constant of 119.3 was used, and for PanOptix, an A-constant of 119.1 was used.

If the corneal astigmatism was greater than 0.8 D upon preoperative measurements, toric IOL placement was indicated. The Barrett Toric Calculator, provided by the American Society of Cataract and Refractive Surgery, was used to obtain predicted postoperative refraction for both Synergy and PanOptix patients, including a target of emmetropia. For each patient, the surgically induced astigmatism was at 0.1 D based on incision size. IOL power and incision location were included for analysis.

## 3. Results

### 3.1. Patient Demographics

All 140 patients (224 eyes) received either a Synergy IOL or PanOptix IOL (Synergy: 105 eyes, 69 patients; PanOptix: 119 eyes, 71 patients). Upon stratifying for the presence of a toric lens, 31 eyes received a toric lens in the Synergy group, and 34 eyes received a toric lens in the PanOptix group (Table 1). At the time of surgery, the mean age of the Synergy and PanOptix groups were 67 ± 8.56 and 66.83 ± 7.43 years, respectively. Stratification based on hyperopia and myopia did not reveal any trend.

Of the patients included in this study, seven patients from the Synergy group underwent YAG capsulotomy within three months of surgery, and twelve patients from the PanOptix group underwent YAG capsulotomy within the first year (*p* = 0.33). One Synergy patient exchanged their IOL for a monofocal TECNIS IOL, and one PanOptix patient required an IOL rotation with a subsequent PRK enhancement. There were no ruptures of the posterior capsule, endophthalmitis, or any other postoperative complications in any patients. A total of 19 eyes from the Synergy group used the Zeiss IOLMaster 700 for biometry measurements, while all remaining eyes from the Synergy and PanOptix groups used the Lenstar LS 900. All preoperative biometry measures between groups showed no statistically significant differences (Table 1).

### 3.2. Refractive Outcomes

Average MRS at six months postoperatively was better in the Synergy group (0.013 ± 0.44 D) compared to the PanOptix group (0.30 ± 0.31 D) (*p* = 0.02). The difference in MRC between patients who received Synergy or PanOptix toric or non-toric lenses was only significant preoperatively (*p* < 0.001). All other differences between MRS, MRC, and MRSE between Synergy and PanOptix, toric and non-toric lenses at every other time point did not demonstrate statistical significance.

### 3.3. Uncorrected Distance Visual Acuity

No difference between preoperative UDVA and three and six months postoperative UDVA was observed between Synergy and PanOptix lenses (*p* = 0.35, 0.32). Additionally, 97% and 98% of patients with Synergy and PanOptix IOLs, respectively, achieved 20/40 vision or better by six months. Whilst 34% of patients with Synergy IOLs achieved 20/20 vision or better by six months postoperatively, 36% of patients with PanOptix IOLs achieved 20/20 vision or better by this same time point (Figure 1). At three months postoperatively, there were more eyes in the PanOptix group with vision better than 20/40 compared to the Synergy group (*p* = 0.04).

### 3.4. Uncorrected Intermediate Visual Acuity

At three months postoperatively, fewer patients achieved 20/20 vision or better in the Synergy group compared to the PanOptix group, but the difference in average UIVA was not statistically significant (*p* = 0.45). At six months postoperatively, the PanOptix group had a slight advantage over the Synergy group, but the difference was not statistically significant (*p* = 0.27). (Figure 1)

### 3.5. Uncorrected near Visual Acuity

The average UNVA in the Synergy group was 0.12 ± 0.07 logMAR and 0.10 ± 0.07 logMAR at three and six months postoperatively, while the average UNVA in the PanOptix group at these time points was 0.16 ± 0.10 logMAR, and 0.18 ± 0.14 logMAR (*p* = 0.01, 0.002). When stratifying for toric and non-toric Synergy and PanOptix lenses, differences for average UNVA were shown at three and six months postoperatively (*p* = 0.01, 0.03). Relatively more patients achieved 20/20 UNVA or better in the Synergy group compared to the PanOptix group at three and six months postoperatively. (Figure 1) At six months, more eyes in the Synergy group had 20/32 vision or better compared to the PanOptix group (*p* = 0.006), and more eyes in the Synergy group had 20/25 vision or better compared to the PanOptix group (*p* = 0.03).

### 3.6. Corrected Distance Visual Acuity

There was no difference in CDVA preoperatively compared to three and six months postoperatively. The average CDVA at these time points in the Synergy group was 0.01 ± 0.03 logMAR and 0.03 ± 0.05 logMAR, respectively, and the average CDVA at these time points in the PanOptix group was −0.03 ± 0.30 logMAR and 0.01 ± 0.03 logMAR, respectively (*p* = 0.29, 0.20). One-hundred percent of patients in both the Synergy and PanOptix groups achieved 20/40 vision or better by six months postoperatively, and relatively fewer patients achieved 20/20 vision or better in the Synergy group compared to the PanOptix group at six months postoperatively (79% vs. 92%, *p* = 0.19). (Figure 1)

### 3.7. Subjective Measures

Reported night vision disturbances were different at the one-month, three-month, and six months postoperatively (*p* = 0.01, 0.03, 0.02, respectively). While the Synergy group reported more night vision disturbances at one and three months, the PanOptix group reported more night vision disturbances at six months. The Synergy group reported more glare than the PanOptix group at one month postoperatively (*p* = 0.002) (Figure 2).

### 3.8. Vector Analysis

The mean vector of astigmatism is represented by the centroid, and one standard deviation from the centroid is represented by the ellipse. The graph’s rings each represent 1.00 D. The preoperative and postoperative keratometric astigmatism values for the Synergy and PanOptix groups are summarized in Figure 3A,D, respectively.

The preoperative corneal astigmatism centroid in the Synergy group was 0.47 D at 95 ± 1.47 D. The centroid of the postoperative refractive astigmatism was 0.09 D at 172 ± 0.52 D (Figure 3B). The preoperative corneal astigmatism centroid in the PanOptix group was 0.46 D at 101 ± 1.47 D. The centroid of the postoperative refractive astigmatism was 0.19 D at 105 ± 0.51 D (Figure 3E). In the postoperative corneal plane, the postoperative ellipse decreased.

The predictive error was ≤1.00 D in 93% of patients in the Synergy group (Figure 3C) and ≤1.00 D in 94% of patients in the Panoptix group (Figure 3F). Effective prediction of toric calculators used in the Synergy and PanOptix groups was indicated by data from double-angle plot analysis for the postoperative refractive astigmatism prediction error.

## 4. Discussion

Multifocal IOLs in the past have shown superior clinical outcomes when compared to monofocal IOLs. While monofocal IOLs provide excellent results with distance vision, they are associated with an increased spectacle dependence following surgery [13,14]. This study demonstrated that the Synergy IOL provided better visual outcomes for UNVA than the PanOptix IOL at the three and six-month postoperative time intervals. These findings are supported by other studies which showed improved postoperative UNVA with Synergy IOLs at three months postoperatively [4,8,9]. Although the PanOptix IOL has been shown to have superior UNVA compared to EDOF IOLs such as the Symfony IOL, Synergy’s superior UNVA performance may be explained by Synergy’s combined multifocal and EDOF properties” [4,6,8,9,10].

The lack of statistically significant differences found for UDVA between the Synergy and PanOptix IOLs at any time point is consistent with other studies and reflects the theoretical benefits of multifocal IOLs, which are meant to maintain distance vision [9,14]. Similarly, there was no significant difference in comparing CDVA between the IOL groups. Although this is contrary to Ferreira et al., which showed that Synergy had better CDVA for close vergence demands, the present study did not assess such demands and therefore was not observed [10]. However, there were significantly more eyes in the PanOptix group with superior UDVA performance at three months postoperatively. Given that this trend did not continue at the three and six-month intervals, it is likely that this finding is inconsequential. Ferreira et al. also found no significant difference in UIVA between Synergy and PanOptix [10].

It has previously been demonstrated that multifocal IOLs increase the incidence of photic phenomena compared to monofocal IOLs [13]. In this study, it was found that earlier on, more Synergy patients reported photic phenomena than PanOptix patients, which later diminished. Specifically, Synergy patients reported more glare at one month postoperatively and more night vision disturbances at one and three months postoperatively (*p* = 0.002, 0.01, 0.03, respectively). However, PanOptix patients did report more night vision disturbances than Synergy patients at the six-month interval (*p* = 0.02). It is possible that patients at the six-month time interval experienced neuroadaptation, accounting for the reversal between the Synergy and PanOptix groups [15]. However, the spherical base curve of the diffractive surface of the Synergy IOL may help explain the relatively increased photic phenomena reported by Synergy patients since spherical surfaces have been associated with high-order aberrations and straylight [16,17]. The spherical base curve of the diffractive surface of the Synergy IOL may also explain enhanced mesopic performance and image contrast in these lenses when compared to PanOptix IOLs, as reported by Dick et al., and provide insight into the increased reported frequency of photic phenomena for Synergy patients [4,5,17].

One factor which can affect visual outcomes is the malrotation of toric IOLs. Of all the patients in this study, only one patient from the PanOptix group required IOL repositioning. Overall, based on the visual outcomes, vector analysis, and only one eye requiring surgical repositioning, both IOLs demonstrated high rotational stability. When compared to non-toric patients, toric patients had good refractive outcomes overall. This is supported by a study by Rementería-Capelo et al., which showed no statistical differences in visual acuity, visual function, or refractive outcomes between non-toric and toric lenses [18]. Further research is required to better ascertain the rotational stability of the Synergy and PanOptix IOLs to evaluate which IOL would be most advantageous in treating patients with astigmatism.

When comparing the results of the current study to the FDA clinical trial results (Figure 4), our results either met or slightly fell short of the visual outcomes described by the FDA. When comparing the current study to other studies published in the literature on the Synergy and PanOptix IOLs, the data varied (Appendix A and Appendix B). This study only recorded monocular visual outcomes, which made comparisons difficult between studies that had recorded primarily binocular visual outcomes. Further, binocular measurements have been suggested to provide a more accurate representation of visual performance [19].

While the sample size used was comparable to other studies in the literature, the possibility for error may be increased due to a relatively limited sample size, which decreased as the postoperative interval increased. Furthermore, intermediate visual acuity was not measured at each postoperative visit, with distance and near vision being the primary measurements obtained.

Another limitation was the absence of grading photic phenomena, specifically glare, halo, and night vision disturbances, as we did not use subjective questionnaires and objective assessments for these outcomes. This retrospective study also did not have complete data on contrast sensitivity, spectacle independence, patient satisfaction, or subjective quality of vision, which are all areas for future investigation. Though the presence of posterior capsular opacification (PCO) could be confounding, all patients with significant PCO underwent YAG capsulotomy, so it is unlikely a major limiting factor. Two biometry systems were used and were not equally distributed between groups, which may have led to differences in refractive outcomes.

In addition, this study assessed visual outcomes using the PanOptix TFAT00 model, while the FDA study used the TFNT00 model. The major difference between the two models is the blue light filtering chromophore present only in the TFNT00 model. Despite this difference, we were still able to compare our visual outcomes to the FDA trial, as the presence of chromophore should not impact measurements or differences in visual acuity [20].

## 5. Conclusions

In conclusion, the Synergy and PanOptix IOLs both provided satisfactory results in near and distance vision, but the Synergy IOL demonstrated superiority with near vision at three months and six months postoperatively. Although both IOLs appear to have good visual outcomes at near, intermediate, and distance vision, it is necessary to counsel patients before surgery regarding the possibility of photic phenomena, including glare and halo. While both IOLs are designed differently, they appear to equally benefit patients who desire spectacle independence by preserving near, intermediate, and distance vision.

## Figures and Tables

**Figure 1 jcm-11-07374-f001:**
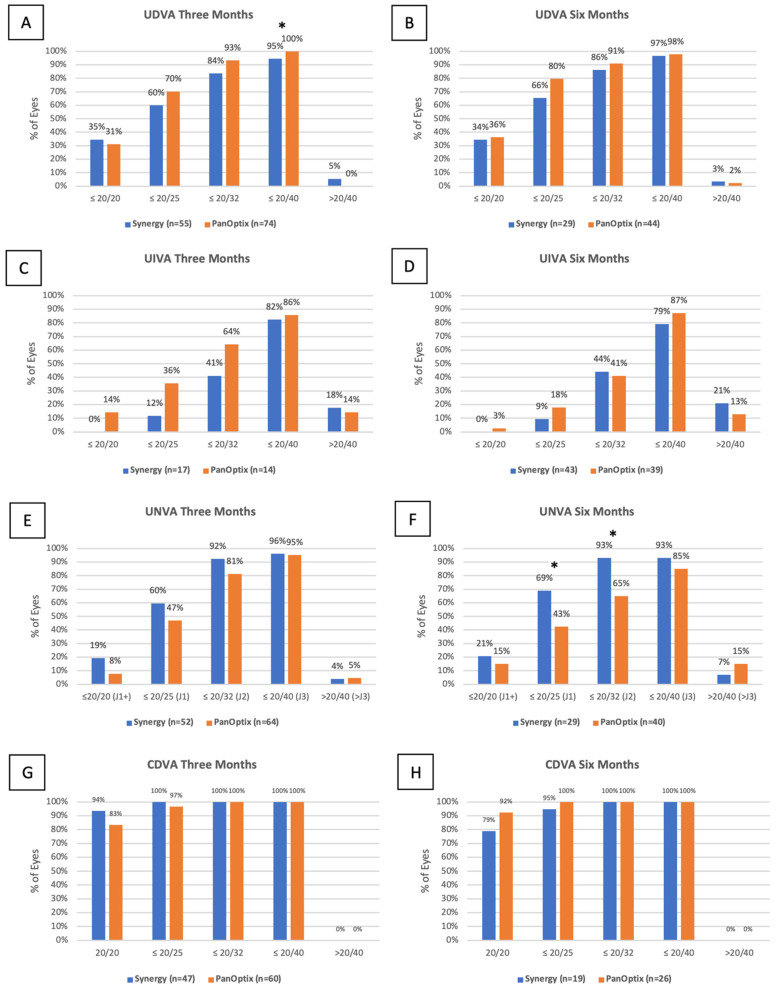
Distribution of monocular UDVA (**A**,**B**), UIVA (**C**,**D**), UNVA (**E**,**F**) and CDVA (**G**,**H**) visual acuity at three months and six months in the TECNIS Synergy and AcrySof IQ PanOptix groups. (* = statistically significant interval).

**Figure 2 jcm-11-07374-f002:**
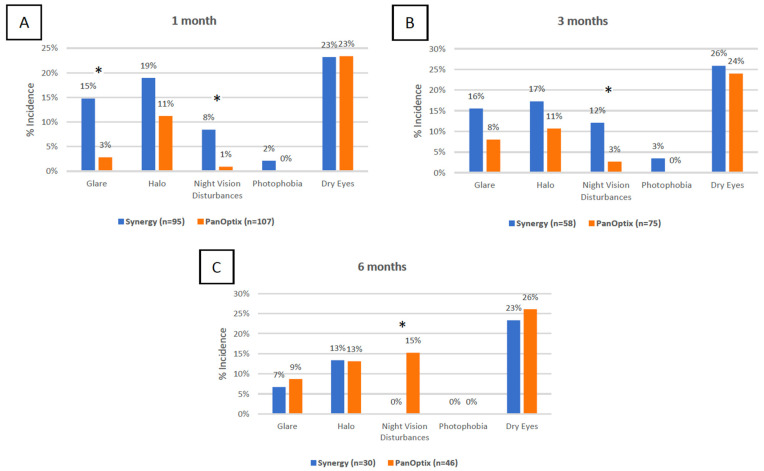
Distribution of secondary visual outcomes at the one-month: (**A**), three months (**B**), and six months (**C**) time intervals in the TECNIS Synergy and AcrySof IQ PanOptix groups. (* = statistically significant interval).

**Figure 3 jcm-11-07374-f003:**
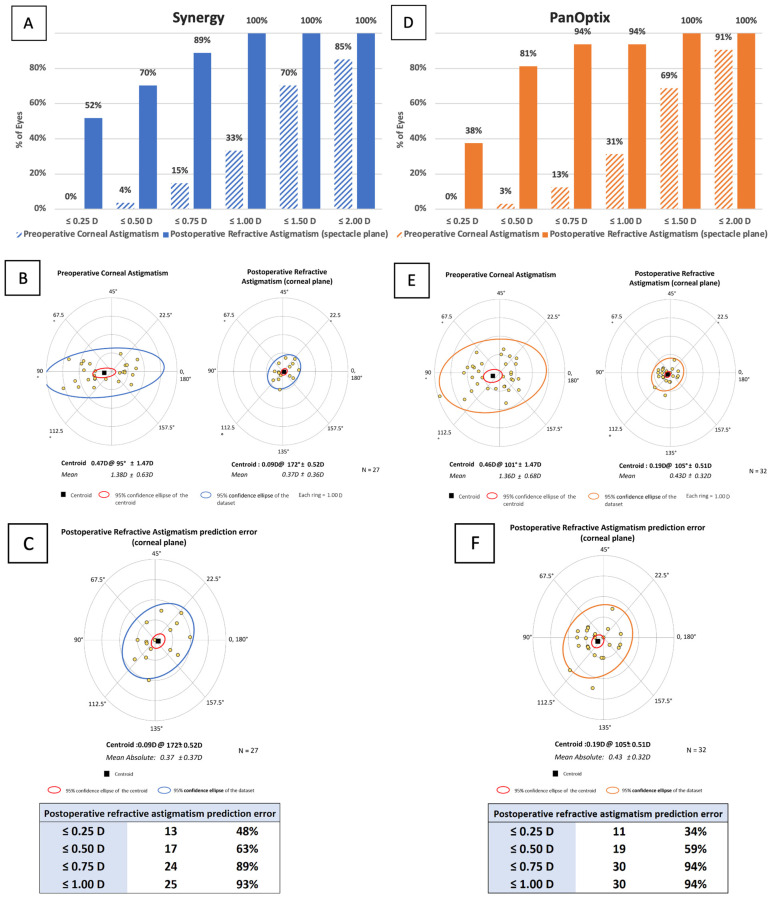
Vector analysis for the TECNIS Synergy group and the AcrySof IQ PanOptix group: (**A**,**D**) Preoperative corneal astigmatism and postoperative refractive astigmatism cumulative histogram measuring magnitude; (**B**,**E**) Preoperative corneal astigmatism and postoperative refractive astigmatism double-angle plots showing the centroid and standard deviation; (**C**,**F**) Refractive astigmatism prediction error double-angle plot, postoperative refractive astigmatism prediction error values.

**Figure 4 jcm-11-07374-f004:**
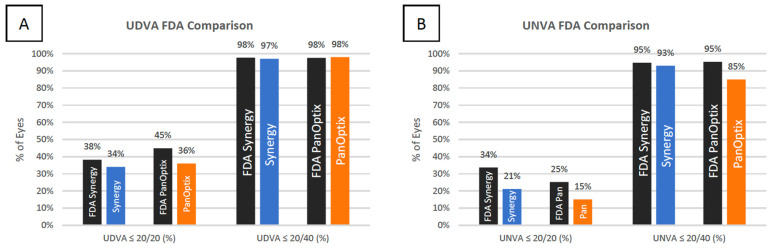
Comparison of the distribution of FDA values with the current study values for monocular UDVA (**A**) and UNVA (**B**) for the TECNIS Synergy and AcrySof IQ PanOptix IOLs.

**Table 1 jcm-11-07374-t001:** Preoperative Patient Demographics.

Preoperative Parameters	Synergy	PanOptix	Non-Toric Synergy	Non-Toric PanOptix	Toric Synergy	Toric PanOptix	*p* Value
**Eyes (*n*)**	105	119	74	85	31	34	-
**Sex (*n*, %)**							
Males	27 (39.1)	38 (53.5)	19 (38.0)	30 (54.5)	12 (52.2)	12 (46.2)	-
Females	42 (61.0)	33 (46.5)	31 (62.0)	25 (45.5)	11 (47.8)	14 (53.8)	
**Age (years)**							
Mean ± SD	67 ± 8.56	66.83 ± 7.43	66.47 ± 7.57	67.27 ± 7.30	68.29 ± 10.58	65.74 ± 7.73	0.83
Range	(39, 83)	(45, 81)	(39, 78)	(53, 81)	(47, 83)	(45, 77)	
**IOP (mmHg)**							
Mean ± SD	13.9 ± 3.1	14.7 ± 3.6	13.9 ± 3.1	15.1 ± 3.7	13.7 ± 3.1	13.7 ± 3.3	0.07
Range	(4, 21)	(8, 30)	(4, 21)	(8, 30)	(6, 20)	(9, 26)	
**AL (mm)**							
Mean ± SD	23.99 ± 1.15	23.75 ± 1.07	23.94 ± 1.23	23.85 ± 0.92	24.10 ± 0.97	23.51 ± 1.34	0.18
Range	(20.63, 27.22)	(20.29, 27.16)	(20.63, 27.22)	(22.02, 27.16)	(22.63, 26.39)	(20.29, 27.0)	
**ACD (mm)**							
Mean ± SD	3.25 ± 0.36	3.19 ± 0.35	3.24 ± 0.35	3.23 ± 0.35	3.27 ± 0.40	3.09 ± 0.33	0.24
Range	(2.24, 4.11)	(2.44, 4.43)	(2.3, 4.02)	(2.61, 4.43)	(2.24, 4.11)	(2.44, 3.97)	
**AD (mm)**							
Mean ± SD	2.71 ± 0.38	2.64 ± 0.33	2.71 ± 0.36	2.67 ± 0.33	2.70 ± 0.42	2.56 ± 0.32	0.3
Range	(1.7, 3.61)	(1.91, 3.56)	(1.8, 3.45)	(2.07, 3.56)	(1.7, 3.61)	(1.91, 3.42)	
**K_m_ (D)**							
Mean ± SD	43.67 ± 1.26	43.72 ± 1.40	43.79 ± 1.25	43.54 ± 1.38	43.40 ± 1.27	44.15 ± 1.38	0.21
Range	(39.1, 47.0)	(39.8, 47.2)	(40.95, 47.0)	(39.8, 46.6)	(39.1, 45.75)	(41.15, 47.2)	
**LT (mm)**							
Mean ± SD	4.44 ± 3.76	4.50 ± 0.40	4.43 ± 0.39	4.49 ± 0.39	4.46 ± 0.35	4.53 ± 0.44	0.69
Range	(3.09, 5.21)	(2.98, 5.75)	(3.09, 5.16)	(2.98, 5.51)	(3.74, 5.21)	(3.89, 5.75)	
**WTW (mm)**							
Mean ± SD	12.09 ± 0.47	12.01 ± 0.42	12.13 ± 0.42	12.01 ± 0.43	12.01 ± 0.56	12.03 ± 0.40	0.43
Range	(10.7, 13.62)	(10.98, 12.84)	(10.7, 13.16)	(10.98, 12.84)	(10.79, 13.62)	(11.14, 12.76)	
**Sphere (D)**							
Mean ± SD	−0.46 ± 2.92	−0.16 ± 2.86	−0.57 ± 3.04	0.03 ± 2.57	−0.17 ± 2.65	−0.62 ± 3.49	0.73
Range	(−8.25, 4.75)	(−9.75, 7.75)	(−8.25, 4.75)	(−9.75, 3.0)	(−7.25, 3.75)	(−8.25, 7.75)	
**SEQ (D)**							
Mean ± SD	−0.88 ± 2.98	−0.61 ± 2.99	−0.90 ± 3.11	−0.30 ± 2.64	−0.76 ± 2.65	−1.39 ± 3.65	0.53
Range	(−8.63, 3.88)	(−10.5, 6.88)	(−8.63, 3.88)	(−10.5, 2.75)	(−7.75, 2.38)	(−9.38, 6.88)	

Abbreviations: M/F = male/female; IOP = intraocular pressure; AL = Axial Length; ACD = Anterior Chamber Depth; AD = Aqueous Depth; K_m_ = Mean Keratometry; LT = Lens Thickness; WTW = White-to-White; SEQ = Spherical Equivalent.

## Data Availability

The data presented in this study are available on request from the corresponding author. The data are not publicly available due to concern for maintaining patient privacy.

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
