# Peer review of "Comparative Visual Outcome Analysis of a Diffractive Multifocal Intraocular Lens and a New Diffractive Multifocal Lens with Extended Depth of Focus"

_jcm, 2022, doi:10.3390/jcm11247374_

Round 1
Reviewer 1 Report
This manuscript compared the visual outcomes of a diffractive multifocal IOL (PanOptix) and a diffractive multifocal lens with extended depth of focus IOLs (Synergy). The study demonstrated that Synergy IOL provided better visual outcomes for UNVA than the PanOptix IOL, and reported more early photic phenomena than PanOptix patients, which later diminished. This article is well written and add new data to the current literature about Synergy IOL.
Besides the visual acuity and vision disturbance, readers may also be interested in the contrast sensitivity, spectacle independency, quality of vision or satisfaction of patients of two IOLs.
Author Response
Reviewer 1
- Besides the visual acuity and vision disturbance, readers may also be interested in the contrast sensitivity, spectacle independency, quality of vision or satisfaction of patients of two IOLs.
Thank you for your suggestion. Unfortunately, we do not have complete data for these metrics. We have added this to our limitations section on lines 301-303:
“This retrospective study also did not have complete data on contrast sensitivity, spectacle independence, patient satisfaction, or subjective quality of vision, which are all areas for future investigation.”
Reviewer 2 Report
No informations are presented on retinal conditions before and after the surgery.
visual differences might be due to a bad retinal conditions before the surgery.
neither this point is considered as exclusion criteria.
yag laser occurrence in my experience is too low the AA must consider that point.
Author Response
Reviewer 2
- No informations are presented on retinal conditions before and after the surgery. visual differences might be due to a bad retinal conditions before the surgery. neither this point is considered as exclusion criteria.
Thank you for your comments. In our methods section on line 77, we mentioned that we excluded from this study anyone with previous retinal disease. In our revised manuscript, we have edited this to say that we excluded patients with “retinal abnormalities”.
- yag laser occurrence in my experience is too low the AA must consider that point.
After reviewing our data set, we confirmed that we had seven Synergy patients and twelve PanOptix patients receive a YAG capsulotomy as stated on lines 146-147. We have now edited our statement to reflect the differing timelines of both IOLs. Given the recent release of the Synergy IOL, our patients who received this lens did so more recently than the PanOptix IOL patients. Thus, several Synergy patients had not yet reached a year since having their surgery. We added more precise language to the revised manuscript on lines 146-148:
“Of the patients included in this study, seven patients from the Synergy group underwent YAG capsulotomy within three months of surgery and twelve patients from the PanOptix group underwent YAG capsulotomy within the first year (p=0.33).
Reviewer 3 Report
The topic “Comparative Visual Outcome Analysis of a Diffractive Multi-2 focal Intraocular Lens and a New Diffractive Multifocal Lens 3 with Extended Depth of Focus” is principally interesting, while the manuscript provides significant weakness:
1. Methodology:
- how many patients received one of the both target IOLs in the same period and were NOT included and for what reasons?
- which preoperative refractive error was accepted?
- did both surgeons implant both models or one the Synergy and one the PanOptix?
- Was the target refraction in all cases zero?
- Which chart was used for conversion of Jaeger to Snellen?
- How many eyes were excluded because of perioperative complications and why (lines 78-79)? Safety is a relevant factor even in the highly successful cataract surgery.
- If follow up was six months, one-year capsulotomy data are beyond the study period. In neither IOL type, early capsulotomy (before 12 months) would be a quality indicator for IOL selection 8% is rather much. Was any capsulotomy done because of its optic relevance? What is significant PCO (line301)? à censor all VA results for patients with PCO …
- Surgically induced astigmatism estimated to be 0.1dpt based on what? Incision size? Personal experience or ASCRS data basis?
- at which astigmatic cut off were toric lenses implanted?
- how was dry eye disease diagnosed/defined?
- were conservative-free topic NSAIDs and corticosteroids used? If not, postop dry eye syndrome findings cannot meaningfully be interpreted.
2. Statistics:
- Both eyes were included for 56 patients which is a relevant bias factor, at least for patient satisfaction, since patients not undergoing second-eye surgery may have been the unhappy ones: Either use – what would methodologically correct – include only the first-operated eye or remove all patient reported outcomes.
3. Results:
- The reported discriminating p-value of p<0.05 is a generally accepted standard. This implies that anything above p0.05 is NOT significant which means, three is NO difference. Why not stick to the own definitions, namely given this is a retrospective analysis (i.e. lines 176-80 and lines 198-200).
- Two biometry systems were used. Were these equally distributed between the groups? Were differences in the refractive outcomes linked to the 2 machines excluded?
- if a p-value is reported, the adjective “significant” is not required given a cut off has been defined.
- The statement in lines 187-9 is not in line with fig 1 A+B. Pls correct and replace the unprecise term “vision” by UDVA or UNVA if this is what you wish to state.
- Assuming that the sample was not 100% complete at all time points, the number of observations (n1/n2) needs to be reported along with the figure at least for the significant findings (see statement in lines 293-5).
- to which comparison is the p-value in table 1 associated? Columns 1+2?
- please explain/interpret the discordant findings in figs 1C and 2C.
4. Discussion:
- Line 243: Not true, see fig 1E.
- contradictory findings in the 1st par deserve to be discussed
- lines 245-7 compared to which IOL?
- the statement in lines 255-7 indicates that the authors assume that their study was not sufficiently powered to support several of their findings, which explains the aforementioned inconsistent or discordant findings. I think at this point, the clear weakness of this study deserves to be discussed. Given the inconsistency of findings, any conclusion must be carefully considered. The last sentence of the conclusion is well-chosen, that no relevant difference between the two IOLs was uncovered. At a maximum, Synergy may do better in UNVA while PanOptix patients report less night vision disturbance.
Taken together, the methodology used here cannot be assessed, so that the robustness of findings has to be questioned.
Author Response
- Methodology:
- how many patients received one of the both target IOLs in the same period and were NOT
included and for what reasons?
We included all patients who received these target IOLs on our data sheet if they had records of post op visits.
- which preoperative refractive error was accepted?
After consulting our data sheet, we did not implant the Synergy or PanOptix IOL in eyes that were myopic more than 10 diopters or astigmatic more than 3 diopters. We have added these details to lines 79-81:
“Preoperative refractive error did not exceed 10 diopters for myopic correction or 3 diopters for astigmatic correction.”
- did both surgeons implant both models or one the Synergy and one the PanOptix?
Thank you for the question. Both surgeons in this study implanted both the Synergy and PanOptix IOLs. We have added this to line 72:
“The surgeons in this study implanted both types of IOLs.”
- Was the target refraction in all cases zero?
Yes. We mention that we include a target of emmetropia on line 130.
- Which chart was used for conversion of Jaeger to Snellen?
The chart used was from the following source:
- Schwiegerling, Field Guide to Visual and Ophthalmic Optics, SPIE Press, Bellingham, WA (2004).
We have added this source to our reference list.
- How many eyes were excluded because of perioperative complications and why (lines 78-
79)? Safety is a relevant factor even in the highly successful cataract surgery.
See lines 150-154:
“One Synergy patient exchanged their IOL for a monofocal TECNIS IOL and one PanOptix patient required an IOL rotation with a subsequent PRK enhancement. There were no ruptures of the posterior capsule, endophthalmitis, or any other post operative complications in any patients. ”
These patients were both included in our study.
neither IOL type, early capsulotomy (before 12 months) would be a quality indicator for
IOL selection 8% is rather much. Was any capsulotomy done because of its optic
relevance? What is significant PCO (line301)? → censor all VA results for patients with
PCO …
Thank you for the question. This was addressed in our previous revision.
After reviewing our data set, we confirmed that we had seven Synergy patients and twelve PanOptix patients receive a YAG capsulotomy as stated on lines 146-147. We have now edited our statement to reflect the differing timelines of both IOLs. Given the recent release of the Synergy IOL, our patients who received this lens did so more recently than the PanOptix IOL patients. Thus, several Synergy patients had not yet reached a year since having their surgery. We added more precise language to the revised manuscript on lines 148-150:
“Of the patients included in this study, seven patients from the Synergy group underwent YAG capsulotomy within three months of surgery and twelve patients from the PanOptix group underwent YAG capsulotomy within the first year (p=0.33).”
- Surgically induced astigmatism estimated to be 0.1dpt based on what? Incision size?
Personal experience or ASCRS data basis?
Surgically induced astigmatism was estimated to be 0.1 dpt based on the incision size. We have added this to lines 138-139:
“For each patient, the surgically induced astigmatism was at 0.1 D based on incision size.”
- at which astigmatic cut off were toric lenses implanted?
On lines 134-135, we state “if corneal astigmatism was greater than 0.8 D upon preoperative measurements, toric IOL placement was indicated.”
- how was dry eye disease diagnosed/defined?
Dry eye disease was diagnosed/defined based on the subjective reporting of the patient in their history. We have added this clarification to lines 99-102:
“Secondary outcomes included patient-reported photic phenomena, specifically glare, halo, night vision disturbances, photophobia, and dryness, which were recorded based on the subjective reporting of the patient at each postoperative visit (one-, three-, and six-months).”
- were conservative-free topic NSAIDs and corticosteroids used? If not, postop dry eye
syndrome findings cannot meaningfully be interpreted.
Preservative-free topical NSAIDs and corticosteroids were not used.
- Statistics:
- Both eyes were included for 56 patients which is a relevant bias factor, at least for patient
satisfaction, since patients not undergoing second-eye surgery may have been the unhappy
ones: Either use – what would methodologically correct – include only the first-operated
eye or remove all patient reported outcomes.
Thank you for your comment. This is certainly a limitation of this study, but we believe the data is still valuable.
- Results:
- The reported discriminating p-value of p<0.05 is a generally accepted standard. This
implies that anything above p0.05 is NOT significant which means, three is NO difference.
Why not stick to the own definitions, namely given this is a retrospective analysis (i.e. lines
176-80 and lines 198-200).
Thank you for the suggestion. We have removed all redundant “significant” labels since we have defined p < 0.05 as significant.
- Two biometry systems were used. Were these equally distributed between the groups?
Were differences in the refractive outcomes linked to the 2 machines excluded?
Thank you for bringing this to our attention. We have added the following to lines 154-156:
“19 eyes from the Synergy group used the Zeiss IOLMaster 700 for biometry measurements, while all remaining eyes from the Synergy and PanOptix groups used the Lenstar LS 900.”
We have also added the following to our limitations section on lines 310-312:
“Two biometry systems were used and were not equally distributed between groups, which may have led to differences in refractive outcomes.”
- if a p-value is reported, the adjective “significant” is not required given a cut off has been
defined.
Thank you for the suggestion. We have removed all redundant “significant” labels since we have defined p < 0.05 as significant.
- The statement in lines 187-9 is not in line with fig 1 A+B. Pls correct and replace the unprecise term “vision” by UDVA or UNVA if this is what you wish to state.
Thank you for the comment. We have corrected this and made more precise terms in Lines 192-193:
“Relatively more patients achieved 20/20 UNVA or better in the Synergy group compared to the PanOptix group at three- and six-months postoperatively.”
- Assuming that the sample was not 100% complete at all time points, the number of
observations (n1/n2) needs to be reported along with the figure at least for the significant
findings (see statement in lines 293-5).
Thank you for the suggestion. We mentioned on lines 142-143 of our paper exactly how many eyes for each IOL. We believe this is sufficient to address this issue.
- to which comparison is the p-value in table 1 associated? Columns 1+2?
The p-value was calculated using an ANOVA test, so the comparison included all six columns. This is mentioned in line 105:
“…continuous variables were statistically analyzed using a one-way ANOVA F-test…”
- please explain/interpret the discordant findings in figs 1C and 2C.
We did not identify discordant findings in these figures. Further clarification would be needed.
- Discussion:
- Line 243: Not true, see fig 1E.
Thank you for bringing this to our attention. While in figure 1E it does not show significant differences at three months, we did calculate significant differences in average UNVA logMAR values for three and six months postoperatively, as stated in lines 187-191.
- contradictory findings in the 1st par deserve to be discussed - lines 245-7 compared to which IOL?
Thank you for the comment. We have edited this to say the following on lines 250-253:
“Although the PanOptix IOL has been shown to have superior UNVA compared to EDOF IOLs such as the Symfony IOL, Synergy’s superior UNVA performance may be explained by Synergy’s combined multifocal and EDOF properties.”
- the statement in lines 255-7 indicates that the authors assume that their study was not
sufficiently powered to support several of their findings, which explains the aforementioned
inconsistent or discordant findings. I think at this point, the clear weakness of this study
deserves to be discussed. Given the inconsistency of findings, any conclusion must be
carefully considered. The last sentence of the conclusion is well-chosen, that no relevant
difference between the two IOLs was uncovered. At a maximum, Synergy may do better in
UNVA while PanOptix patients report less night vision disturbance.
Thank you for the comment. We do discuss at length the limitations of this study in the discussion, and we have added further limitations and corrected other items suggested by the reviewers. We agree that our conclusions should be carefully considered given these limitations.

Round 2
Reviewer 2 Report
the AA did not addressed completely my suggestion:
1. what does it mean retinal abnormalities? if patient has drusen or RPE deficits was included or not in the study,
2. in my opinion posterior capsulotomy occurence rate is too low (at one year 13.6%) while in literature in normal cataract extraction is around 30%, may be due to a patient drop out during the follow up. the AA must discuss this point with higher accuracy
Author Response
- What does it mean retinal abnormalities? If patient has drusen or RPE deficits was included or not in the study?
Thank you for your response. All patients who are candidates for multifocal IOLs routinely undergo OCT. Patients are screened for abnormal macular findings such as drusen and RPE changes.
- In my opinion posterior capsulotomy occurence rate is too low (at one year 13.6%) while in literature in normal cataract extraction is around 30%, may be due to a patient drop out during the follow up. the AA must discuss this point with higher accuracy.
The occurrence rate of yag capsulotomy varies in the literature. On our review, we found many studies with similar YAG occurrence rates to our study, which is about 10%. The yag rate may increase with time. As mentioned in our revision on lines 149-150 we followed the Synergy patients for at least 3 months and the PanOptix patients for one year.
